# Examining the role of information integration in the continued influence effect using an event segmentation approach

**Jasmyne A. Sanderson** *, **Simon Farrell, Ullrich K. H. Ecker**

School of Psychological Science, University of Western Australia, Perth, WA, Australia

* jasmyne.sanderson@research.uwa.edu.au

## Abstract

Misinformation regarding the cause of an event often continues to influence an individual's event-related reasoning, even after they have received a retraction. This is known as the continued influence effect (CIE). Dominant theoretical models of the CIE have suggested the effect arises primarily from failures to retrieve the correction. However, recent research has implicated information integration and memory updating processes in the CIE. As a behavioural test of integration, we applied an event segmentation approach to the CIE paradigm. Event segmentation theory suggests that incoming information is parsed into distinct events separated by event boundaries, which can have implications for memory. As such, when an individual encodes an event report that contains a retraction, the presence of event boundaries should impair retraction integration and memory updating, resulting in an enhanced CIE. Experiments 1 and 2 employed spatial event segmentation boundaries in an attempt to manipulate the ease with which a retraction can be integrated into a participant's mental event model. While Experiment 1 showed no impact of an event boundary, Experiment 2 yielded evidence that an event boundary resulted in a *reduced* CIE. To the extent that this finding reflects enhanced retrieval of the retraction relative to the misinformation, it is more in line with retrieval accounts of the CIE.

## Introduction

Misinformation regarding the cause of an event can continue to influence people's event-related reasoning even if it has been retracted in a clear and credible manner by the issuing of a correction. This is known as the continued influence effect [CIE; 1–8]. The CIE has been demonstrated under controlled laboratory environments; for example, in a study by Johnson and Seifert [2], participants were given a news report detailing a fictitious warehouse fire. Participants were originally told that the fire was caused by negligent storage of volatile materials, but this was later retracted. Despite acknowledging the retraction, when answering questions about the fire, participants continued to rely on the initial incorrect information. Arguably, the CIE has also played a role in a number of real-world misinformation cases such as the MMR-autism scare, where despite strong evidence to the contrary, misinformation linking the MMR vaccine to autism had significant and lasting influence [3, 9–12].

**Data Availability Statement:** Materials are provided in the Supplement. Data are available on the Open Science Framework website at https://osf.io/qrpjb/ (doi:10.17605/OSF.IO/QRPJB; Experiment

1) and https://osf.io/sxh2w/ (doi:10.17605/OSF.IO/
SXH2W; Experiment 2).

**Funding:** This research was supported by an
Australian Government Research Training Program
Scholarship and a Bruce and Betty Green
Postgraduate Research Scholarship to JS. The
funders had no role in study design, data collection
and analysis, decision to publish, or preparation of
the manuscript.

**Competing interests:** The authors have declared
that no competing interests exist.

Previous research has focused primarily on the role that memory retrieval plays for the CIE. Specifically, the CIE has been assumed to arise from selective retrieval of the misinformation, or failure to retrieve the correction. According to this view, memory representations compete for activation, and the CIE occurs if the misinformation is retrieved but its correction is not [13–15]. A competing explanation stems from a mental-model account. This account proposes that people build mental models of events as they unfold [16–18]. When new information relevant to the event is encountered, the mental model of the event needs to be updated. It has been argued that this updating process is particularly effortful if the new information features a correction that supersedes earlier information, because a correction requires the removal of old outdated information (i.e., the misinformation) and the encoding and integration of the new information to form a coherent updated model that reflects the current state of affairs [19–22]. Therefore, if a correction is not sufficiently integrated during model updating, people may continue to rely on the outdated misinformation in their inferential reasoning.

Recent neuropsychological research has supported the mental-model account, indicating that the CIE may arise in part due to a failure in the integration and updating of information in memory [23]. Using functional magnetic resonance imaging, Gordon et al. [23] found differing neural activity for the encoding of retractions compared to non-retractions in regions involved in information integration (i.e., the precuneus and posterior cingulate gyrus), whereas no retraction-vs.-non-retraction difference was found during retrieval. This was taken to suggest that the CIE may arise due to a failure to integrate a retraction into a person's mental model. However, a follow-up study failed to find neural differences at encoding; instead the results implicated a failure of retrieval processes [24].

It is therefore still unclear exactly what role integration and updating failure play in the CIE. This study expanded on previous research by investigating information integration and updating using a behavioural approach. Specifically, this study sought to manipulate the ease with which a retraction can be integrated with misinformation through use of event segmentation boundaries.

## Event segmentation effects on integration and memory

As people go about their everyday life, they experience a continuous stream of incoming information, and yet this information is often remembered as distinct events or episodes in memory. There are a number of prominent models and theories of event cognition that can explain how this occurs [25–27]. In particular, event segmentation theory [EST; 28–30] and more recently the event horizon model [26] propose that individuals segment incoming information into discrete, meaningful events—a segment of activity that is perceived to have a defined beginning and end—which are separated by event boundaries. This segmentation process is thought to be automatic and triggered by contextual change (i.e., a change in one or more dimensions of the contextual environment such as spatial location). In other words, if context is perceived as constant, incoming information will continue to be interpreted as part of the same event; however, when there is a deviation in context, an event boundary is created, and thus two separate event representations [31]. This is consistent with the finding that individuals often identify event boundaries in text narratives and films at points where the features of a situation change [e.g., a character changes location; 29, 32–35].

According to EST, events are represented in memory within an *event model*—a mental representation of the structure and content of the current event—which aids comprehension and is used to guide predictions about future behaviour and perceptual input. Event models are updated at event boundaries due to the increase in prediction error that occurs when aspects of the event change. EST argues for a global updating process, in which following an

event boundary, the old event model is abandoned and a new model is created. The new model is then actively maintained in memory until the next boundary is encountered [36]. In this way, event segmentation acts as a control process that regulates and organises information in memory through updating at event boundaries [37]. This then has implications for the accessibility of information after a boundary has been crossed [34].

It follows that event segmentation can impact learning and memory [28, 30]. In particular, information from a specific event is better recalled while the event is still ongoing, as opposed to after the event, when an event boundary lies between the encoding of the information and its retrieval. Previous research has supported the effect of event boundaries on information accessibility using text narratives, film, and virtual-reality (VR) environments. These studies typically find that crossing an event boundary decreases the accessibility of information encoded prior to the boundary, as reflected in increased response times and reduced response accuracy, as well as slower reading times for boundary compared to non-boundary sentences [17, 28, 35, 37–46]. These effects occur independently of pure temporal parameters such as the absolute amount of time since encoding or the temporal distinctiveness of the to-be-recalled information [47].

## Spatial event segmentation

Changes in spatial context—which is generally an important dimension of memory representations [e.g., 26]—have been investigated as triggers of event segmentation [35, 48]. Shifts in spatial context (i.e., a change in location) may act as event-boundary markers that can subsequently disrupt cognitive processing and memory retrieval [26, 39, 49, see 50 for a review]. Consistent with this, Rinck and Bower [41] had participants memorise the layout of a building with specific objects located within each room, before reading a narrative about a character moving through the building. After the character moved to a new room, readers were slower at recognising objects from previous rooms. Similar results have been found for interactive environments.

The most prominent example of this is the location updating effect, which refers to the decline in memory that results when an agent moves from one location to another [e.g., passing through a doorway; e.g., 39, 51, 52]. In a series of experiments conducted by Radvansky and colleagues [39, 51, 52], participants navigated a series of rooms while picking up and setting down different objects in each room. During this, participants' recognition memory for the objects was tested. The critical factor of interest was whether or not participants shifted to a new room between encoding and test. These studies found that memory for objects was less accurate and slower following a shift in location compared to when there was no shift. In other words, the accessibility of information from the previous room decreased after a spatial event boundary had been crossed. This finding was even more pronounced when participants moved through multiple rooms, therefore crossing multiple event boundaries [51]. The effect occurs in both VR [38, 39, 51, 52] and physical laboratory environments [51], as well as in people's imagination [53]. It has been found when word pairs were used instead of objects [52], when the rooms were separated by transparent "glass" walls allowing participants to preview the next location [54], when recall was tested as opposed to recognition [55], and for both younger and older adults [56]. Furthermore, the effect does not appear to be simply due to context-dependent memory which would suggest that returning to the location where information was initially learned would improve recall [e.g., 57, 58], as memory was affected even when individuals were tested in the original room of encoding [51].

In summary, consistent with event segmentation theory, it appears that the presence of spatial event boundaries in either text narratives, film, or interactive environments influences the

ease with which information is integrated and later accessed, with disruptions to memory occurring for information encoded prior to a boundary.

## The current study

The current study investigated the effect of spatial event segmentation boundaries on information integration and updating in a CIE paradigm. According to the mental-model account and event segmentation theory, a spatial event boundary between the encoding of the misinformation and retraction should disrupt retraction integration and model updating. To illustrate, the work of Kendeou [21, 22, also see 59–61] has shown that memory updating is facilitated when the initial misinformation and its subsequent correction are co-activated in memory and a conflict detected. This should be made less likely by an event boundary occurring between the two pieces of information, as encountering the correction may fail to trigger successful retrieval of the earlier misinformation. In turn, this could hinder integration and mental-model revision, thus resulting in increased misinformation reliance.

Across two experiments, misinformation and retractions were presented with or without a spatial event boundary between their encoding. To employ the spatial event boundary, participants either moved physically (Experiment 1) or virtually (Experiment 2) between spatial contexts. Physical/virtual spatial context changes were chosen so as to accommodate the event report format typically used in CIE studies, which do not lend themselves to narrative spatial shifts. For example, in contrast to the story-based narratives typically used in event segmentation studies, event reports are by nature less linear in terms of structure and less confined to a particular context/person (e.g., written in third person with input from multiple sources), which makes it difficult to implement event boundaries. Both experiments included inference questions to assess misinformation reliance.

## Experiment 1

The aim of Experiment 1 was to investigate the role of information integration and updating in the CIE. Specifically, it aimed to examine the effect of a physical spatial event boundary on the integration of misinformation and its retraction, applying an event segmentation approach to the CIE paradigm. We manipulated the presence versus absence of a spatial event boundary, via a shift in spatial context, between misinformation and retraction encoding. Participants either read both the misinformation and retraction in the same spatial context (i.e., the same room; no-shift condition) or in two different contexts by shifting locations to a new room before reading the retraction (shift condition). Participants then answered a series of inferential reasoning questions (in a third room) to determine the effect of the boundary on retraction integration and subsequent misinformation reliance.

There were two hypotheses: The main hypothesis was that integration would be facilitated in the no-shift condition—when misinformation and retraction were encoded in the same spatial context—and that this would reduce subsequent misinformation reliance, relative to the shift condition in which there was a spatial event boundary between misinformation and retraction encoding. A secondary hypothesis was that a retraction (in either shift or no-shift conditions) would reduce reliance on misinformation compared to a no-retraction control, without entirely eliminating its influence, replicating previous work [e.g., 15, 62, 63].

### Method

Experiment 1 used a between-subjects design contrasting no-shift and shift conditions. Due to the resource-intensive nature of testing, it was decided to just implement these two conditions, while obtaining data from a no-retraction control condition from a separate sample of online

participants, simply to obtain a rough baseline for comparison as per the secondary hypothesis. The dependent variable was participants' inference score derived from their responses to post-manipulation inference questions.

## Participants

The original target sample size was derived using a-priori power analysis [using G*Power 3; 64, also see 65] which suggested that to detect a difference between no-shift and shift conditions inference scores in a *t*-test, an effect of $f = .20$ ($\alpha = .05$; 1-$\beta = .80$) would require a sample of 200 participants. The effect size of $f = .20$ was chosen based on Brysbaert [65] who recommends $f = .20$ as a default effect size for power analyses in psychological research. In addition, previous research [41] found an effect of $f = .23$ when comparing no-boundary and boundary conditions using an ANOVA. Due to difficulties in recruiting participants, it was recognised that a sequential Bayesian approach would be appropriate to balance the resource-intensive nature of testing against the evidence gained. Accordingly, the data were subjected to Bayesian sequential testing after testing half the number of planned participants, and committing to Bayesian statistics for inference beyond that point [66, 67]. As described below, preliminary results provided clear evidence for the null, and the experiment was stopped so as to not waste resources.

A total of $N = 112$ first-year undergraduate students from the University of Western Australia participated in exchange for course credit. The total sample comprised 77 females and 35 males; mean age was $M = 20.83$ years ($SD = 5.87$), ranging from 17 to 49 years. Participants were randomly assigned to either the no-shift or shift condition (with the constraint of roughly equal participant numbers; $n = 56$ in no-shift condition; $n = 56$ in shift condition).

A separate sample of $n = 56$ participants were recruited on Amazon MTurk [68] for the no-retraction control condition. Participants were U.S. residents; the sample comprised 26 females and 30 males; mean age was $M = 37.23$ years ($SD = 10.61$), ranging from 21 to 64 years. Participants were reimbursed US$1.80 through MTurk.

## Materials

**Event reports.** Four event reports detailing fictitious scenarios (e.g., an emergency airplane landing) were used in the present study (although participants received only one out of the pool of four). These reports were based on those from Ecker et al. [62]. Each report comprised 12 individual messages and was separated into two parts—Part 1 (messages 1–6) contained the critical information (e.g., that the emergency landing was due to extreme weather conditions), and Part 2 (messages 7–12) contained either the retraction (e.g., that initial reports were incorrect and the landing was due to mechanical failure) or a neutral filler message (e.g., that the airline released a statement) in the no-retraction control condition. It was decided to not explicitly refer to the misinformation within the retraction as previous research [62] has found that explicit retractions were more effective in reducing misinformation reliance, presumably by facilitating conflict detection and model updating. As such, referring explicitly back to the misinformation may facilitate integration, which may reduce the impact of the event boundary manipulation. See S1 File for the full event reports.

**Distractor tasks.** Two distractor tasks were used. The first was a word finding task, which was completed between the two parts of the event report. This was used to achieve comparable time intervals between misinformation and retraction encoding across conditions. To this end, the distractor task duration was 2 min in the no-shift condition (and the control condition) but only 1 min in the shift-condition (to allow time for the move from one room to the other). The second distractor task was a word fluency task completed after the event report.

Six letters were presented successively on the screen for 30 s each (i.e., 3 min total), and participants had to type as many words as possible starting with each letter.

**Questionnaires.** There was one questionnaire specific to each report. Each questionnaire consisted of 11 questions that related to the relevant event report: an open-ended inference question, four memory questions and six rating scale inference questions. The questionnaires are provided in the S1 File.

The four memory questions assessed memory for details mentioned in the report to determine that participants were paying attention during encoding. These questions were presented in multiple-choice format with four response options (e.g., *What airport did the airplane land at?–a. Portland; b. Denver; c. Orlando; d. Seattle*). Each question assessed a different part of the report (i.e., before and after the critical information, before and after the retraction/filler statement).

Seven inference questions were used to assess participants' reliance on the critical information. The first question was an open-ended question that asked participants to briefly summarise the report. The remaining six were statements relating to the critical information (e.g., *The US guidelines for flying in bad weather should be reviewed*). Agreement with each statement was rated on an 11-point Likert scale ranging from "*Completely disagree*" (0) to "*Completely agree*" (10). One of these statements was negatively coded.

There was an additional question at the end of the experiment asking participants whether they had put in a reasonable effort (participants answering "no" were excluded from analysis; see below for details).

**Rooms.** The event reports were read in one or two rooms, depending on condition. The rooms and respective survey versions (see below) were set up to be as different as possible (see Fig 1). Room A was decorated with four landscape posters on adjacent walls, dim lighting via a lamp, an air freshener, items on the desk (e.g., plant, scent diffusor, tissues), and an office chair with wheels. The Room A survey was presented on a small square monitor (16-inch LG Flatron L1919S); it used a black background and the text was white Georgia size 12 font. Room B had plain white walls, overhead lighting, no items on the desk, and a padded steel frame chair without wheels. The Room B survey was presented on a large rectangular monitor (21.5-inch ViewSonic VX2252mh); it used a white background with blue Arial size 16 font. Room C was a communal lab space with four workstations set up, each with a computer monitor and office chair with wheels; one wall contained a bookshelf and another a window.

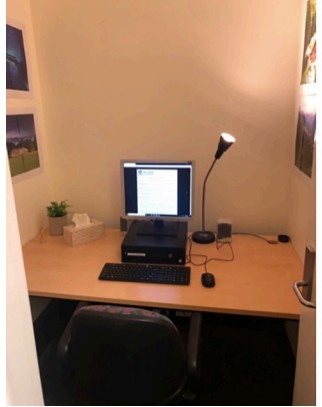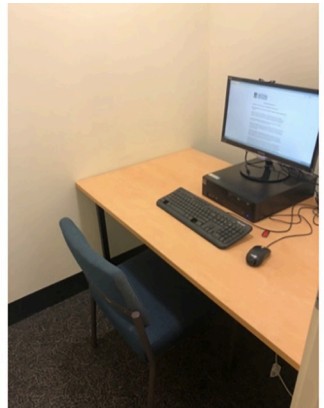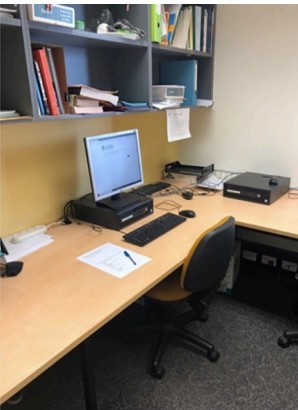

**Fig 1. Rooms used in Experiment 1.** Left: Room A; middle: Room B; right: Room C.

## Procedure

Ethics approval was granted by the University of Western Australia's Human Research Ethics Office. Participants received an approved information sheet and provided informed consent prior to participating for this and all other experiments. Participants read one event report (out of the pool of four). The report was presented one message at a time on a computer screen, delivered via Qualtrics software (Qualtrics, Provo, UT, USA). The rooms and event report topics were counterbalanced across participants.

In the no-shift condition, participants read both Part 1 and Part 2 of the event report in the same room (Room A or B). The word finding distractor task was completed for 2 minutes in between the two parts of the report in the same room. In the shift-condition, participants read the first part of the event report in one room (Room A or B) and then moved to a second room (Room B or A respectively) to complete the word finding distractor task for 1 minute and read the second part of the report. All participants, regardless of condition, then moved to Room C, where they completed the word fluency distractor task for 3 minutes and answered the relevant pen-and-paper questionnaire (see Fig 2). Participants were then fully debriefed. The experiment took approximately 15 minutes. Apart from the fact that participants completed the experiment online (and thus did not move locations during the survey), the procedure in the control condition was identical.

## Scoring

**Memory scores.** Memory scores were calculated from responses to the four multiple-choice memory questions. Correct responses were given a score of 1 and incorrect responses received a score of 0, resulting in a possible maximum memory score of four.

**Inference scores.** Inference scores were calculated from responses to the open-ended question and the six rating-scale items. The negatively worded items were reverse-scored. Responses to the open-ended question were coded by the experimenter for recall of the critical information, the retraction, and the provided alternative explanation. A score of 1 was given

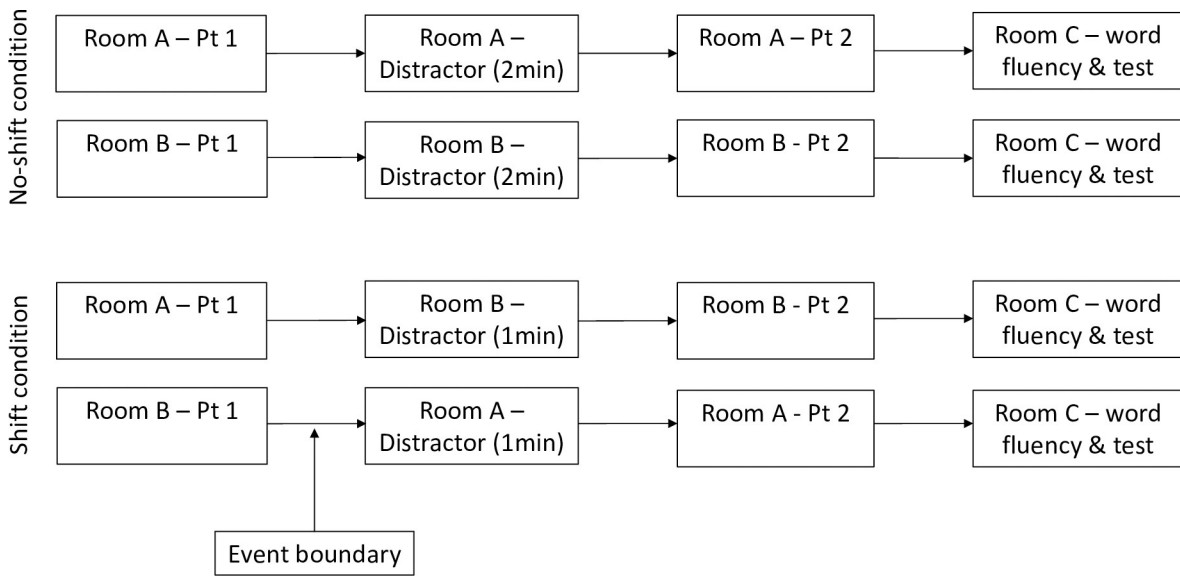

**Fig 2. Schematic depiction of Experiment 1's no-shift and shift conditions procedure.** Note: Pt 1, part one of the event report; Pt 2, part two of the event report.

when the critical information, retraction, or alternative was recalled and 0 otherwise. Thus, participants were able to receive a score of 1 on all three scores (e.g., recall of the critical information: "it was stated the emergency landing was due to bad weather", the retraction: "as it turned out, it was not actually due to the weather", and the alternative: "it was due to a mechanical fault"). The no-retraction control condition was not coded for recall of the retraction or alternative. The coded scores were then converted into an inference score of 0 or 10 to maintain the same scaling as the rating scale inference questions: If participants recalled the retraction and/or the alternative, they received a score of 0 (indicating no misinformation reliance) irrespective of whether they also recalled the critical information; if participants recalled only the critical information without the retraction or alternative, they received a score of 10 (indicating misinformation reliance). A total inference score was then calculated as the average of the seven inference questions; higher inference scores indicate greater misinformation reliance.

## Results

One participant in the control condition failed all memory questions; their data were excluded. Additionally, one participant in the no-shift condition was excluded who indicated that their data should not be used due to lack of effort. Finally, one participant in the shift condition was excluded due to failure to complete the experiment. Thus, the final sample was $N = 165$ ($n = 55$ per condition).

All Bayesian analyses were conducted in JASP, using default priors (fixed-effects scale parameter $r = 0.5$; random-effects scale parameter $r = 1$). Bayes factors ($BF$) represent the relative evidence for one model over another, given the data. Any $BF$ can be expressed as either $BF_{10}$, which quantifies support for the alternative hypothesis, or $BF_{01}$, which quantifies support for the null hypothesis. A $BF$ in the range 1–3 (1/3-1) provides anecdotal evidence, a $BF$ of 3–10 (1/10-1/3) provides substantial evidence, a $BF$ of 10–30 (1/30-1/10) provides strong evidence, a $BF$ of 30–100 (1/100-1/30) provides very strong evidence, and a $BF > 100$ ($< 1/100$) constitutes extreme evidence [69].

### Memory scores

The mean memory scores across the four event scenarios and the three conditions are shown in Table 1.

The mean memory scores were comparable across control ($_C$), no-shift ($_{NS}$), and shift ($_S$) conditions; mean scores were $M_C = 3.26$ ($SE_C = .11$), $M_{NS} = 3.44$ ($SE_{NS} = .09$), and $M_S = 3.46$ ($SE_S = .10$). A Bayesian independent samples $t$-test provided substantial evidence against a difference between no-shift and shift conditions, $t(108) = -.14$, $BF_{01} = 4.91$.

**Inference scores.**   Mean inference scores were $M_C = 6.50$ ($SE_C = .27$), $M_{NS} = 4.00$ ($SE_{NS} = .32$), and $M_S = 3.57$ ($SE_S = .25$). The mean inference scores across the four event scenarios and the three conditions are shown in Table 2.

**Table 1. Descriptive statistics for memory scores across scenarios and conditions.**

| | Airplane Landing | | Bushfire | | Water Source | | Nightclub | |
|---|---|---|---|---|---|---|---|---|
| | *M* | *SE* | *M* | *SE* | *M* | *SE* | *M* | *SE* |
| Control | 2.80 | .20 | 3.53 | .19 | 3.23 | .23 | 3.50 | .20 |
| No-shift | 3.43 | .17 | 3.62 | .14 | 2.92 | .24 | 3.73 | .12 |
| Shift | 3.64 | .13 | 3.29 | .22 | 3.00 | .20 | 3.86 | .14 |

**Table 2. Descriptive statistics for inference scores across scenarios and conditions.**

|  | Min | Max | *M* | *SE* |
|---|---|---|---|---|
| Airplane Landing |  |  |  |  |
| Control | 0.86 | 9.57 | 5.89 | 0.53 |
| No-shift | 3.29 | 7.57 | 5.70 | 0.26 |
| Shift | 2.00 | 7.43 | 4.58 | 0.45 |
| Bushfire |  |  |  |  |
| Control | 3.29 | 8.71 | 5.43 | 0.41 |
| No-shift | 0.00 | 5.86 | 2.26 | 0.58 |
| Shift | 1.43 | 5.57 | 3.69 | 0.35 |
| Water Source |  |  |  |  |
| Control | 3.29 | 10.00 | 7.07 | 0.55 |
| No-shift | 1.86 | 8.29 | 5.01 | 0.56 |
| Shift | 1.43 | 7.29 | 3.70 | 0.49 |
| Nightclub |  |  |  |  |
| Control | 5.43 | 9.86 | 7.99 | 0.37 |
| No-shift | 0.00 | 8.14 | 3.03 | 0.65 |
| Shift | 0.00 | 6.86 | 2.32 | 0.51 |

Numerically, both no-shift and shift conditions had lower inference scores than the no-retraction control, suggesting some effect of the retraction. Two Bayesian one-sample *t*-tests determined that inference scores in both no-shift and shift conditions were substantially different from zero, $ts(54) \geq 12.36$, $BF_{10} \geq 1.68e{+}17$, indicating the presence of a CIE. We acknowledge that zero may not be an appropriate baseline when using rating scales, but use zero in the absence of any other obvious baseline. A Bayesian independent samples *t*-test (with default priors: fixed-effects scale parameter $r = 0.5$; random-effects scale parameter $r = 1$) comparing no-shift and shift conditions yielded substantial evidence against a main effect of condition on participants' reliance misinformation in their inferential reasoning, $t(108) = 1.05$, $BF_{01} = 3.02$ (though numerically the no-shift condition had a higher inference score than the shift condition). In addition, a one-tailed test (specifying $H_1$ as NS < S) was run to quantify the evidence against the directional hypothesis. This provided strong evidence against $H_1$, $t(108) = 1.05$, $BF_{01} = 9.39$.

## Discussion

To investigate the potential role of information integration in the CIE, Experiment 1 examined whether the presence of a spatial event boundary would impair the integration of a retraction. It was predicted that integration would be facilitated when misinformation and retraction were encoded in the same spatial context, and that this would reduce misinformation reliance relative to a condition with a spatial event boundary between misinformation and retraction encoding. The observed data indicated that a retraction in both no-shift and shift conditions reduced reliance on misinformation (without entirely eliminating it) compared to a no-retraction control condition, consistent with prior research [15, 62, 63]. However, contrary to predictions, a spatial event boundary did not have a substantial effect on the integration of a retraction, at least not to the extent that the effect was measurable in participants' misinformation reliance. While numerically there was an effect, it was in the direction opposite of that predicted, with greater reliance on misinformation in the no-shift condition.

This finding is inconsistent with the EST assumption that event boundaries induced through spatial context shifts disrupt access to information from the preceding event. Possible

explanations for why Experiment 1 failed to find evidence for an effect of a spatial event boundary relate to (1) the differences in cognitive processing relative to previous EST studies, (2) the causal structure of the event reports used, and (3) the spatial context manipulation itself.

Firstly, Experiment 1 did not measure retraction integration or retrieval directly. Instead, it used a CIE paradigm, which measures different cognitive processes than the typical paradigms used in EST research. As far as we are aware, this is the first study to investigate the impact of event boundaries in a CIE paradigm. In previous EST research, the effect of event segmentation boundaries has been demonstrated on memory retrieval directly. Based on this prior research, we assumed that segmentation boundaries would reduce mnemonic accessibility of the misinformation, which would in turn impair information integration, resulting in stronger continued influence of the misinformation on inferential reasoning. Therefore, there were two additional processing steps (i.e., integration and continued influence) that had not been previously considered by EST studies. As such, one explanation for the inconsistent results may be that the boundary affected mnemonic accessibility in line with previous research, without this impacting on information integration and/or inferential reasoning downstream. It is also possible that the reasoning measures typically applied in CIE studies are not sensitive enough to detect potential boundary effects.

Secondly, the reports typically used in the EST literature are narratives with a linear timeline (e.g., Jan did this, she then did this, she then did that. . .). The event reports used in Experiment 1 as well as previous CIE research are by nature less linear (e.g., the events described have time gaps or reversals) and have an underlying causal structure (e.g., X caused Y. . .actually it was Z that caused Y), which may have negated the effect of the boundary. The event horizon model suggests that individuals monitor causal information in events, and that causal connections can act as a way of linking events, which may thus influence whether an event is interpreted as being a part of the same or different event model [70–72]. That is, causally related information is more likely than unrelated information to be interpreted as belonging to the same event, irrespective of the underlying temporal structure of the report [35, 70, 71]. Therefore, the causal structure of the event reports may have influenced whether the spatial boundary manipulation had an effect on the integration of the retraction. In addition, it might be speculated that the inclusion of response alternatives in the memory questions may have facilitated general recall of the event, thereby indirectly reducing the impact of the boundary manipulation on inferential reasoning.

Thirdly, it may be that the rooms were not as contextually different as was required to find spatial event segmentation effects. While every attempt was made to make the rooms different in a perceptually salient manner, there were practicality constraints, as the rooms used for encoding were both testing booths with similar layouts (e.g., small rooms containing a desk against one wall) within a larger lab space. It may have been that rather than interpreting the two rooms as two different spatial locations, participants may have interpreted them as being elements of one larger location (i.e., the lab), resulting in only one event model. Stronger manipulations, for example through use of a VR environment, may allow a greater distinction between the spatial locations. Alternatively, it may be the case that the rooms were sufficiently different but that the spatial context manipulation lacked task relevance. Some studies have failed to produce spatial boundary effects with narrative texts unless participants were specifically instructed to monitor spatial context [73–77]. As such, it has been suggested that if spatial context information is not in an individual's attentional focus (i.e., emphasised by or important to the text or task), it may be less likely to be noticed and encoded [39, 71, 77, 78]. This suggests that people may not necessarily construct event models based on spatial location unless it is made salient or functionally important to the task [74]. In addition, the spatial

context change occurred outside the narrative environment (as opposed to within the narrative; e.g., a character changing locations) and it may be that in order for participants to perceive the boundary as relevant to the narrative there needed to be both narrative and extra-narrative spatial shifts [see 79, for a similar argument].

## Experiment 2

Experiment 2 sought to extend Experiment 1 by using a VR environment to better differentiate between the locations, and to make the spatial shift more central to the memoranda. Participants played the role of a detective who read clues with the expectation that the clues would be needed for a later task in the story setting. The clues related to a fictional crime, and bore on the likely culpability of a main suspect. The misinformation implicated the main suspect, with other clues not bearing directly on the guilt of that suspect. Participants either read both the misinformation and retraction in the same spatial context (i.e., a living room depicting a crime scene; no-shift condition) or in two different contexts by shifting locations to a police station foyer before reading the retraction (shift condition). Participants then answered a series of inferential reasoning questions (in a third room; a police interrogation room) to determine the effect of the boundary on retraction integration and subsequent misinformation reliance.

## Method

Experiment 2 used a between-subjects design contrasting no-shift and shift conditions. Due to the resource-intensive nature of testing, it was decided to implement these two core conditions only. The dependent variables were participants' inference scores derived from their responses to the post-manipulation inference questions, and a "charge" measure based on a question asking participants which suspect they would charge. The design and analysis plan were pre-registered (https://osf.io/kg73n).

### Participants

We anticipated that a sample size of 100 would be feasible, given practicality constraints; this was the pre-registered sample size. A total of 127 students from the University of Western Australia and community members participated. No demographics were recorded. Participants were randomly assigned to either the no-shift or shift condition (with the constraint of roughly equal cell numbers; $n = 64$ in no-shift condition; $n = 63$ in shift condition).

### Apparatus

**VR equipment.**   We used a HTC Vive Pro headset, a controller, and two tripod base stations. The task was programmed in Unity and run via SteamVR on a Gigabyte Sabre 15 Laptop, with a resolution of 1920 x 1080 pixels.

### Materials

**Event report.**   The event report was displayed as a series of messages that became available at certain points during the participants' exploration of the VR environment (see details below). The report detailed a house robbery, and was based loosely on Johnson and Seifert's [2] jewellery theft narrative. The report was separated into two parts. Part 1 contained a general explanatory paragraph setting the scene (i.e., the participant was told they are a detective who has arrived at a crime scene to investigate a burglary), followed by eight clues about the crime (e.g., a blue thread potentially originating from the resident's scarf), which included the critical information (i.e., a clue that implicated a particular suspect, viz. the homeowner's son

Evan; e.g., a bookmaker's betting slip and reports from the family indicate Evan has a gambling problem and has accumulated large debts). Part 2 of the report contained eight retraction statements presented as emails, which retracted both non-critical information (e.g., clarifying that the thread did not belong to the homeowner's scarf) and the critical information (i.e., several sources confirmed that the son was out of town). Retractions of non-critical information were included in order to not overly highlight the retraction of the critical information, and to maintain consistency with previous CIE research, which tends to present retractions not in isolation but in the context of other filler information [e.g., 80, 81]. See S1 File for the full event report.

**Questionnaire.**   The questionnaire consisted of eleven questions that related to the event report: four inference questions, one charge question, and six memory questions. All questions are provided in the S1 File.

The six memory questions assessed memory for details mentioned in the report, to determine that participants were paying attention during encoding. These questions were presented in multiple-choice format with four response options (e.g., *What colour was the button*?—*a. Blue; b. Green; c. Red; d. Yellow).*

The four inference questions were used to assess participants' reliance on the critical information. These questions were presented as statements relating to the critical information (e.g., *How likely are the family to be angry with their son*?). Agreement with each statement was rated on an 11-point Likert scale ranging from 0 to 10.

The charge question was a multiple-choice question asking participants who they were going to charge (i.e., *Who will you charge*?—*a. Evan [son]; b. H. Brown [known offender]; c. Nobody [insufficient evidence]).* This was included as an exploratory measure only.

**VR rooms.**   Participants stood in a virtual environment, which was one of three locations: the crime scene, the police station foyer, and the interrogation room (see Fig 3). The crime scene location was designed as a living room (i.e., warm lighting, lounge, sliding doors), which was markedly different from the police station foyer (i.e., bright lights, officer sitting behind a glass divider). The interrogation room was a dimly lit room and contained a metal desk. In the VR environment, a controller was visible (its position and orientation linked to that of the physical controller held by the participant) and emitted a laser to show where it was pointing. The bottom trigger of the controller was used to record a response.

## Procedure

Participants were tested individually in the lab. They first received general task instructions (including cautions about use of VR equipment) and a cover story, before putting on the VR set. Participants remained standing for the entirety of the task with their back to a wall. The experimenter adjusted the VR headset and the inter-display distance until the participant reported that the environment was in focus (not blurry) and comfortable. Participants first completed a familiarisation phase in the living room, where they practiced looking around and using the controller in the VR environment and responding to prompts.

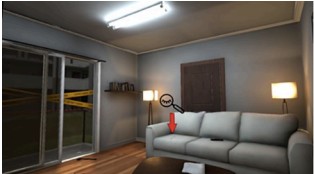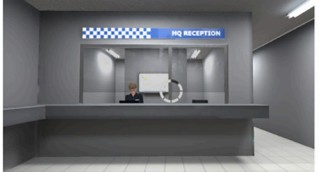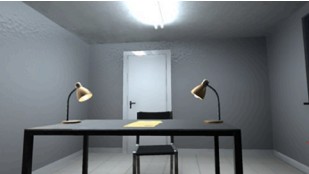

**Fig 3. Rooms used in Experiment 2.** Left: Crime Scene; middle: Foyer; right: Interrogation Room.

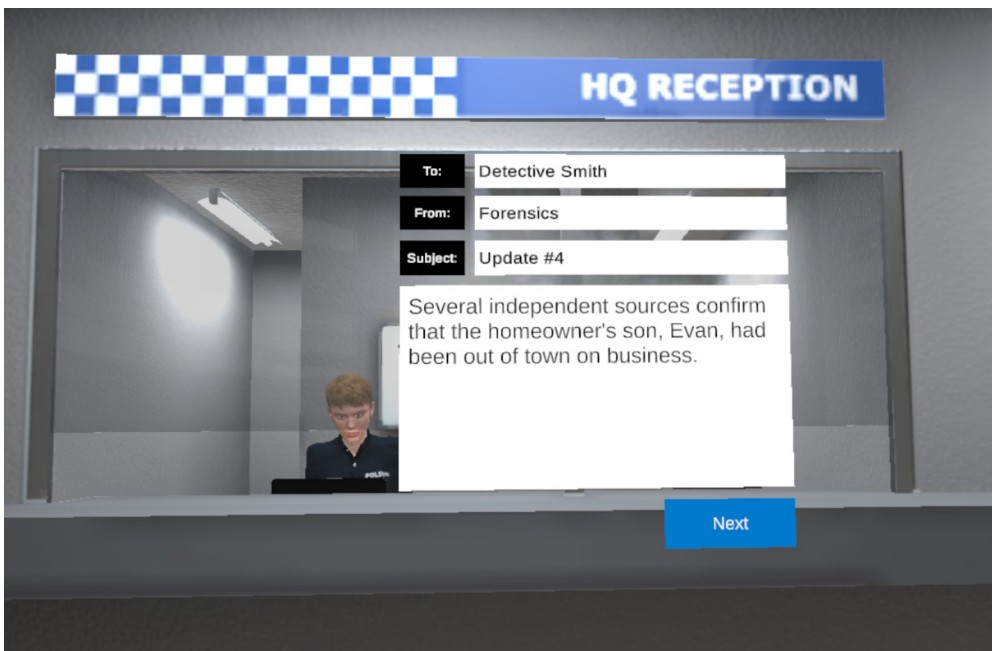

**Fig 4. Example of Experiment 2's retraction email in the shift condition.**

Participants then received a more detailed introduction to the crime scene and explored the crime scene. Participants were able to inspect clues by finding and clicking on a magnifying glass. The magnifying glass for each clue appeared at 10 s intervals; participants were able to find clues before the magnifying glass appeared; however, participants rarely managed to do this given the timing. Participant could inspect the clues for as long as they liked. Clues were queued such that a clue would not appear until the previous clue had disappeared. As a consequence, the clues in the living room in Part 1 were presented in a fixed order.

Once all clues had been provided, the information from Part 2 was presented, which involved retractions of both non-critical and critical information. Each item in Part 2 was presented as an email in the VR environment; an example is shown in Fig 4. There was a 15 s interval between the last clue and the first email. Participants read each email and then moved on to the next email by clicking the 'Next' button.

In the no-shift condition, participants read both Part 1 and Part 2 of the event report at the crime scene and then moved to the police station for a brief period, where they were asked to wait briefly until the interrogation room was available. In the shift-condition, participants read Part 1 at the crime scene and then moved to the police station to read Part 2. Following a 10 s interval, all participants then moved to the interrogation room, where they received the questionnaire. The timing was more or less controlled between conditions (although participants determined the timing of clues and emails by how fast they read and responded).

On completion of the experiment, participants were fully debriefed. The experiment took approximately 15 minutes.

## Scoring

**Memory scores.** Memory scores were calculated from responses to the six multiple-choice memory questions. Correct responses were given a score of 1 and incorrect responses received a score of 0, resulting in a possible maximum memory score of six.

**Inference scores.**   Inference scores were calculated by averaging responses to the four rating-scale items. The negatively worded item was reverse-scored. Higher inference scores indicate greater misinformation reliance, with a maximum possible score of 10.

**Charge scores.**   Charge scores were based on the multiple-choice charge question. Responses were converted into a score of 0 or 10 (to maintain consistent scaling with the rating scale inference items): If participants chose Evan (the son), they received a score of 10 (indicating misinformation reliance); if participants chose nobody or an alternative suspect, they received a score of 0 (indicating no misinformation reliance).

## Results

Based on pre-registered a-priori minimum-performance outlier criteria, two participants with poor memory for the materials (defined as less than 50% correct) were excluded from analysis, yielding a final sample of $N = 125$ ($n = 64$ in no-shift condition; $n = 61$ in shift condition). In line with our pre-registration, analyses were conducted on $N = 100$ (the first 100 completions; $n = 52$ in the no-shift condition; $n = 48$ in the shift condition). However, additional analyses conducted on the full sample ($N = 125$) were comparable and are provided.

### Memory scores

The mean memory scores were comparable across no-shift ($M_{NS} = 5.12$; $SE_{NS} = .13$) and shift ($M_S = 5.06$, $SE_S = .14$) conditions. A Bayesian independent-samples $t$-test provided substantial evidence against a difference between conditions in participants' memory for the report, $t(98) = .28$, $BF_{01} = 4.58$.

With $N = 125$, mean memory scores were $M_{NS} = 5.08$, $SE_{NS} = .12$ and $M_S = 4.98$, $SE_S = .13$. A Bayesian independent-samples $t$-test provided substantial evidence against a difference between conditions, $t(123) = .54$, $BF_{01} = 4.59$.

### Inference scores

Mean inference scores were $M_{NS} = 4.24$ ($SE_{NS} = .22$), and $M_S = 3.43$ ($SE_S = .29$). Two Bayesian one-sample $t$-tests determined that no-shift and shift condition inference scores were substantially different from zero, $ts(47/51) \geq 11.91$, $BF_{10} \geq 3.92e+21$, indicating the presence of a CIE. To test the hypothesis that misinformation reliance would be lower in the no-shift condition compared to the shift condition, an undirected (two-tailed) Bayesian independent-samples $t$-test was run on inference scores. This provided anecdotal evidence for a difference between conditions, $t(98) = 2.25$, $BF_{10} = 1.95$, but in the direction opposite to predictions. In addition, a one-tailed test (specifying $H_1$ as NS < S) was run to quantify the evidence against the directional hypothesis. This provided strong evidence against $H_1$, $t(98) = 2.25$, $BF_{01} = 14.36$, reflecting the finding that contrary to our hypothesis, inference scores were greater in the no-shift compared to the shift condition.

With $N = 125$, mean inference scores were $M_{NS} = 4.29$ ($SE_{NS} = .20$) and $M_S = 3.63$ ($SE_S = .24$). Bayesian one-sample $t$-tests revealed decisive evidence that both conditions were greater than zero, $ts(60/63) \geq 14.85$, $BF_{10} \geq 1.759e+27$. A Bayesian independent-samples $t$-test found inference scores were greater in the no-shift compared to the shift condition, $t(123) = 2.11$, $BF_{01} = 15.36$ (one-tailed).

### Charge scores

The mean charge score across the no-shift and shift conditions was $M_{NS} = 2.12$ ($SE_{NS} = .57$), and $M_S = 1.88$ ($SE_S = .57$). Two Bayesian chi-square tests were conducted to determine if the

conditions charge scores differed from zero (zero indicating no misinformation reliance). Both conditions' charge scores were substantially greater than zero, all $\chi^2 (1) > 17.31$, $BF_{01} <$ 7.109e-4; indicating the presence of a CIE.

To test the hypothesis that the no-shift condition would have lower misinformation reliance than the shift condition, a Bayesian chi-square test (using the default prior: $a = 1$) with condition as the sole factor was conducted on participants' charge scores. There was substantial evidence against a main effect of condition on participants' decision to charge a suspect, $\chi^2 (1) = .09$, $BF_{01} = 4.86$.

With $N = 125$, mean charge scores were $M_{NS} = 2.03$ ($SE_{NS} = .51$), and $M_S = 1.97$ ($SE_S = .51$). Two Bayesian chi-square tests revealed substantial evidence that both conditions were greater than zero, all $\chi^2 (1) > 22.44$, $BF_{01} < 4.684e-5$. A Bayesian chi-square test suggested substantial evidence of no difference between conditions, $\chi^2(1) = .01$, $BF_{01} = 5.63$.

## Discussion

The aim of Experiment 2 was to investigate the effect of a spatial event boundary on the integration of a retraction, and subsequent reliance on misinformation, using a VR environment. From the information integration approach it was predicted that integration would be facilitated in the no-shift condition and that this would reduce misinformation reliance, relative to the shift condition [22, 23, 39, 51, 52]. There appeared to be an effect of a spatial event boundary on participants' inferential reasoning; however, the direction of this effect was opposite to our prediction, with greater misinformation reliance occurring in the no-shift condition compared to the shift condition.

At first glance, this result appears inconsistent with previous event segmentation literature, specifically research on the location updating effect, which has reported that shifting locations (i.e., crossing a spatial event boundary) negatively impacts memory compared to a no-shift condition [e.g., 51, 82]. However, a study by Pettijohn, Thompson, Tamplin, Krawietz, and Radvansky [83] found that event boundaries can occasionally improve memory when they add structure and organisation to the contents in memory, such that the segregation of information into different event models may reduce interference from competing information. Therefore, the boundary between misinformation and retraction may have helped participants in the shift condition retrieve the retraction without interference from the misinformation. In other words, the boundary may have impaired memory for the misinformation relative to memory for the retraction, which in turn reduced accessibility of and reliance on misinformation and thus improved inferential reasoning. The current study's findings are broadly consistent with Experiment 1, inasmuch as Experiment 1 numerically also found greater reliance on misinformation in the no-shift condition and neither experiment yielded evidence for better integration in the no-shift condition.

We also note that due to the no-shift/shift manipulation, the number of spatial boundaries between the retraction and the test varied; specifically, the shift condition had one boundary between retraction and test (misinformation-shift-retraction-shift-test), whereas the no-shift condition had two (misinformation-retraction-shift-shift-test). Therefore, it may be that in the no-shift condition, memory for the retraction was impaired because there were two context shifts between retraction encoding and retrieval at test, compared to only one in the shift condition. From a temporal context view, this would mean that the relative contextual recency of the retraction (vs. misinformation) in the shift condition would reduce reliance on the misinformation [47, 84]. Indeed, previous misinformation research has noted the importance of recency when providing corrections based on the assumption that recent information is more easily retrieved [85–87, also see 88, 89].

## General discussion

The present research aimed to investigate the role of information integration in the continued influence effect (CIE). Specifically, Experiments 1 and 2 examined the effect of a spatial event boundary on the integration of a retraction, using physical and virtual reality environments.

The results of Experiment 1 suggested that a retraction was effective in reducing reliance on misinformation, as reflected in participants' inferential reasoning scores, consistent with previous research [1, 5, 15, 62, 63]. However, while retractions appeared effective, both studies found that they did not completely eliminate reliance on misinformation, providing evidence for a CIE, again in line with previous research [2, 6, 14, 20, 90]. Future research could include a no-misinformation condition in order to formally establish the presence of a CIE.

Across both experiments, we failed to find evidence of a detrimental effect of a spatial event boundary on retraction integration, as reflected in similar inferential reasoning scores between no-shift and shift conditions in Experiment 1 and greater inference scores in the no-shift condition in Experiment 2. We acknowledge that while we can rule out a substantial effect, a small effect may be compatible with the observed results. However, the theoretical and practical relevance of such a small effect are questionable. This is inconsistent with the mental-model account of the CIE, which suggests that people may continue to rely on outdated misinformation in their inferential reasoning if a retraction is not sufficiently integrated during model updating [22, 23]. It was hypothesized that integration of a retraction should be facilitated in the no-shift condition (where the misinformation and its retraction should be part of the same event segment) compared to the shift condition (where the misinformation and its retraction should be part of different event segments), which in turn should result in lower, not greater, post-retraction misinformation reliance. Thus, overall, results provide some evidence against the mental-model account and the role of integration failure in the continued influence effect.

Overall, the results are more in line with the retrieval account of the CIE. This holds that both misinformation and retraction information are stored separately and potentially retrieved at the point of inference, with the reliance on misinformation depending on the ability to retrieve retraction information. This theory provides an explanation for the surprising finding of the lower misinformation reliance in the shift condition in Experiment 2, where the fewer spatial context shifts between the retraction and test may have facilitated retrieval of the retraction relative to the misinformation [also see 24, 81, 91].

Together, these studies have implications for event segmentation theory and the replicability and generalizability of event segmentation effects to a CIE context. In particular, the findings show that event segmentation may only impact specific retrieval processes but not downstream processes such as integration and updating. Future research may wish to investigate the effect of spatial event boundaries within the event reports (e.g., a protagonist moving locations), as opposed to being applied externally to the participant [41, 77]. Future research could also use other types of event boundaries (e.g., shifts in temporal context) [92].

## Supporting information

**S1 File. Supplemental information for Experiments 1 and 2.**
(DOCX)

## Acknowledgments

We thank Ryan Li for Experiment 2's programming, and Charles Hanich for research assistance for Experiment 1 and 2. We also acknowledge the involvement of the PSYC3310 Topic 7 class of 2019 in some testing for Experiment 2.

## Author Contributions

**Conceptualization:** Jasmyne A. Sanderson, Simon Farrell, Ullrich K. H. Ecker.

**Data curation:** Jasmyne A. Sanderson, Simon Farrell, Ullrich K. H. Ecker.

**Formal analysis:** Jasmyne A. Sanderson.

**Investigation:** Jasmyne A. Sanderson.

**Methodology:** Jasmyne A. Sanderson, Simon Farrell, Ullrich K. H. Ecker.

**Project administration:** Jasmyne A. Sanderson, Simon Farrell, Ullrich K. H. Ecker.

**Resources:** Jasmyne A. Sanderson, Simon Farrell, Ullrich K. H. Ecker.

**Software:** Jasmyne A. Sanderson.

**Supervision:** Simon Farrell, Ullrich K. H. Ecker.

**Validation:** Jasmyne A. Sanderson.

**Visualization:** Jasmyne A. Sanderson.

**Writing – original draft:** Jasmyne A. Sanderson.

**Writing – review & editing:** Jasmyne A. Sanderson, Simon Farrell, Ullrich K. H. Ecker.

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
