## [Decision Letter · Decision Letter 0]

1 Mar 2022

PONE-D-21-32228Examining the role of information integration in the continued influence effect using an event segmentation approachPLOS ONE

Dear Dr. Sanderson,

Thank you for submitting your manuscript to PLOS ONE. I have now received comments on your manuscript from two expert reviewers and I have also studied your manuscript myself. Both reviews were positive, with comments that were constructive and clear. After careful consideration, we feel that it has merit but does not fully meet PLOS ONE’s publication criteria as it currently stands. Therefore, we invite you to submit a revised version of the manuscript that addresses the points raised during the review process.

A couple of points stood out for further comment. I agree with Reviewer 2 that results in the supplementary materials section would be better off in the main results section.

More substantively, like Reviewer 1, I also wondered how the effect used in the power analyses of Experiment 1 was chosen. Furthermore, these f effect sizes can be difficult to understand so I wonder if it can be also explained in more concrete terms (see below, and in relation to the specific test(s) used). I highlight this in particular as in the face of null results an important question to ask is how likely was a type 2 error to have occured. While the Bayes Factors did give good support for the null hypotheses, the sample sizes used here for a between-subjects design will likely have limited sensitivity to detect small effect sizes. PLOS-ONE is a journal where our policy is to accept null results where the methods are sound, but in these cases I would normally like to see more consideration of statistical power and discussion of the sensitivity of the design to detect an effect if one were present.

As I understand it, in the case here I note that the main statistical hypothesis tested in the results of Experiment 1 was a difference in conditions in the size of the inference effect - a main effect in a Bayesian ANOVA, with three conditions. However, the hypotheses in the introduction did not straightforwardly map onto this test, with the following hypotheses:

H1: "both no-shift and shift conditions would reduce reliance on misinformation compared to a no-retraction control"

H2: "integration would be facilitated in the no shift condition... relative to the shift condition"

The exact of test of H1 was unclear to me - would this be the average of the no-shift + shift conditions compared with the control condition? H2 seemed to be more in line with a t-test (as in Experiment 2). I suspect that the study had good power to detect a CIE but may have limited power to detect a modulation of the CIE unless the modulation was large. Relating back to the power analysis used as a justification, it mentions "detecting an effect", but not which one, as there are several on offer here (i.e. the CIE effect and effects associated with the 2 hypotheses). This could be clarified, with more detail on the mapping of your hypotheses to tests. Furthermore, with frequentist tests it is straightforward and informative to conduct a sensitivity analyses of your design - this is not conditional on the effect you observed, but what effect sizes you would have good power to detect given the properties of your design and test used. Given the use of Bayesian analysis this might be less straightforward to conduct, but as mentioned above in the face of a null result greater consideration of sensitivity seems necessary, especially as in the discussion it makes reference to numerical differences in the effects across conditions.

We look forward to receiving your revised manuscript.

Kind regards,

Shane Lindsay

Academic Editor

PLOS ONE

Journal Requirements:

Reviewers' comments:

Reviewer's Responses to Questions

**Comments to the Author**

1. Is the manuscript technically sound, and do the data support the conclusions?

Reviewer #1: Partly

Reviewer #2: Yes

2. Has the statistical analysis been performed appropriately and rigorously? 

Reviewer #1: Yes

Reviewer #2: Yes

3. Have the authors made all data underlying the findings in their manuscript fully available?

Reviewer #1: Yes

Reviewer #2: Yes

4. Is the manuscript presented in an intelligible fashion and written in standard English?

Reviewer #1: Yes

Reviewer #2: Yes

5. Review Comments to the Author

Reviewer #1: Reasoning about events can be negatively affect by earlier misinformation even after that information has been corrected. This so-called continued influence effect has been attributed to failures to recollection corrections and to integrative encoding of correct and incorrect information. The present study examined the role of integrative encoding in this effect in the context of event segmentation and memory updating. According to a prominent theory of event comprehension, when people experience event boundaries during misinformation correct, integrative encoding and, thus, memory updating should be disrupted, leading to a greater negative influence of misinformation on subsequent memory accuracy. The presence of spatial event boundaries was manipulated during retractions to examine differences in integrative encoding effects on memory updating. The results from two experiments were mixed as event boundaries had both no effect on memory for corrections (Experiment 1) and reduced the negative effects of misinformation on subsequent memory accuracy (Experiment 2). These findings do not support an integrative encoding account of the continued influence effect and better align with the view that this effect occurs due to retrieval failure.

The topic of the mechanisms underlying the negative proactive effects of misinformation on memory for correct information is quite timely given the high prevalence of misinformation in everyday life. The present study takes a unique approach to illuminating the role of various encoding and retrieval mechanisms in the updating of memories for inaccurate information. I appreciated the creative connection between event segmentation theory and the continued influence effect as well as the innovative experimental procedure. However, the study had some key limitations, such as the ambiguity in results across experiments. On balance, this package seems appropriate for the journal with the caveat that the inconsistency in results across experiments should be highlighted as an impediment for clear interpretation. Below, I describe some concerns and offer specific comments that I hope will help improve the manuscript.

One general issue was that the Introduction did not summary a relevant literature on the role of retrieval during encoding in information integration. The present study appears to examine how disruptions in the contact between misinformation and subsequent retractions impacts memory updating. There are studies from the memory and discourse comprehension literatures that should inspire predictions in the current experiments. Specifically, there is recent memory work from Wahlheim and Zacks (2019), Wahlheim et al. (2020), and Cohen-Sheehy et al. (2021) on this topic. There is also earlier discourse comprehension work from Kintsch and more recent work from Kendeou and colleagues that may be relevant here. The authors may consider how the notion of integrative encoding (information integration) in those studies fits in here. In my view, the findings from these studies showing that memory updating is more effective when recent information elicits retrieval of earlier information also leads to the prediction that reducing the accessibility of earlier information via event boundaries should lead to fewer retrievals and therefore less integrative encoding. This perspective could provide a more complete mechanistic motivation for the hypotheses given in the present study.

Specific Comments:

p. 5, line 99: Do event boundaries decrease the “availability” or “accessibility” of information in memory?

p. 7: It would be great if there was a section here that synthesized the literature review and provided an overarching description of “The Present Study” to give the reader a better understanding of why the three literatures in the Introduction were considered.

pp. 7 and 8, lines 147-152: What was the rationale for using a between-subjects design instead of a within-subjects design? Was there any concern about sampling from multiple populations for the experimental (undergraduates) and control (MTurk workers) conditions? More generally, I could not understand from the description of the experiment what the control condition was intended to assess with respect to the main hypotheses.

p. 8, Participants: How did the authors choose the effect size that was targeted in the power analysis?

p. 9, Event reports: It would be great to have more information about what motivated the structure of the retractions. Specifically, the example here does not include details from the original misinformation statement. Was this done purposefully to decrease the contact between retractions and misinformation to increase the chances that the event boundary manipulation would reduce the accessibility of misinformation? I have a similar query about the materials in Experiment 2.

p. 10, lines 196-201: Do the authors worry that providing response alternatives on the questionnaire had reactive effects on subsequent memory for event details? I have a similar query about the materials in Experiment 2.

p. 10, line 220: I believe the monitor was a 21.5-inch (not 31.5 inch).

p. 11, Procedure: What motivated the decision to administer the final memory test in a unique room (i.e., Room C)? I had the impression that the effects of spatial shift would be most pronounced if the final testing room context matched the retraction encoding context.

p. 12, Memory scores: Are the authors concerned that providing response alternatives on a multiple-choice memory test rescues impaired event details accessibility post-boundary? In other words, could available but less accessible details be made more accessible with the presentation of copy cues during the test?

p. 13, lines 287-290: I was uncertain how the comparison presented here provided support for a continued influence effect. Shouldn’t that require a comparison of misinformation correction items with control items for which no misinformation had appeared? If I have just missed something, it would be great if the authors could explain that here.

p. 14, Discussion: Echoing a concern above, I am not confident that the comparison of experimental groups from one population should be compared directly with a control group from another population. Also, for the possible explanations as to why Experiment 1 failed, the authors may consider my point above that a recognition task could rescue what would otherwise be inaccessible memories of event details following spatial shifts. The authors allude to this possibility in lines 328-329.

pp. 21 and 22: I was confused about the conversion of charge scores into 0 and 10 values. Could the authors have treated this as a 1/0 variable and used some type of logistic regression approach?

General Discussion: This section is a bit shorter than I am accustomed to reading in multi-experiment reports. The authors might consider connecting their findings to the literature more broadly here.

Reviewer #2: In two behavioral studies, Sanderson and colleagues utilized an event segmentation approach to investigate the persistence of misinformation. Given the robustness of the continued influence effect, finding new ways to investigate the phenomenon is a very worthwhile pursuit. Overall, I find the procedure and results easy to follow, and I think the VR detective framing (Experiment 2) is quite clever. Below are a few comments and suggestions which may help further strengthen the paper.

Introduction:

1. As I progressed through the introduction, I was anticipating a study that manipulated event segmentation within the context of the narratives. As such, I was surprised when I came to find out that the manipulation was segmentation of the encoding environment. Further motivation and clarification of that manipulation choice in the introduction will be important.

I should add that although I am generally familiar with the event segmentation literature, I was unfamiliar with the “location updating effect” described on page 6. When I read the description of “…when an agent moves from one location to another…”, I was assuming that the “agent” refers to the protagonist of the narrative. Clarifying that section -- and further explaining how this goes beyond the traditional context-dependent memory effects -- will go a long way towards minimizing potential confusion.

Experiment 1:

2. In coding the open-ended inference question, the authors opted for a broad definition of “no misinformation reliance”, page 12: “If participants recalled the retraction and/or the alternative, they received a score of 0 (indicating no misinformation reliance) irrespective of whether they also recalled the critical information…”

Recent work has shown that there may be an important distinction between different operationalizations of “continued influence” (https://doi.org/10.1186/s41235-021-00335-9). If the open-ended responses were coded differently -- for example, 0 when they recalled only the retraction and/or alternative; 5 points when they recalled retraction and/or alternative AND critical information; 10 = critical information only -- will the patterns look different? This may be a more nuanced way to distinguish among the varying levels of success at integration. This maybe an interesting exploratory analysis, especially since the initial analyses did not yield any effect of segmentation conditions.

3. Minor point: I assume the researchers converted the open-ended question score to 0 or 10 so that the scale would match the close-ended questions? It would be helpful to include an explicit statement about that in the scoring section.

Experiment 2:

4. In Part 2 of the report, retractions of non-critical information were introduced alongside retractions of critical information. Please clarify the rationale for and utility of this design choice.

5. Unless dictated by a prescribed word limit, I strongly encourage the authors to incorporate the Results reported in the Supplement into the main text.

General Discussion:

6. On page 24, the authors wrote, “Both experiments found that a retraction was effective in reducing reliance on misinformation …” Since Experiment 2 did not include a baseline (no retraction) condition, this claim does not apply to Experiment 2. Please rephrase.

7. The paper will benefit from some discussion of the future utility of using the event segmentation approach as a way to understand CIE.

6. PLOS authors have the option to publish the peer review history of their article (what does this mean?). If published, this will include your full peer review and any attached files.

Reviewer #1: No

Reviewer #2: No

---

## [Author Response · Author response to Decision Letter 0]

5 Apr 2022

Please see attached "Response to Reviewers" document.

---

## [Editor Report · Decision Letter 1]

19 May 2022

PONE-D-21-32228R1

Examining the role of information integration in the continued influence effect using an event segmentation approach

PLOS ONE

Dear Dr. Sanderson,

Thank you for submitting your manuscript to PLOS ONE. After careful consideration, we feel that it has merit but does not fully meet PLOS ONE’s publication criteria as it currently stands. Therefore, we invite you to submit a revised version of the manuscript that addresses the points raised during the review process.

I was not able to secure the two previous reviewers to review the re-submission. To expedite the review process, I have considered the revised submission and your responses, and have the following comments below. These are focused on analyses and corresponding conclusions in response to revisions or outstanding issues from the original submission. 

We look forward to receiving your revised manuscript.

Kind regards,

Shane Lindsay

Academic Editor

PLOS ONE

Journal Requirements:

Additional Editor Comments:

1. Myself and Reviewer 1 asked for a justification for the sample size. This was provided in the cover letter but not in the revisions. Could the rationale be provided in the manuscript. If mentioning a previous finding as a justification, could you provide more detail (e.g., exactly which result/test in the Rinck and Bower paper is being used as justification).

2. Further in regard to the justification, this appears based on a frequentist power analyses with a p < .05 decision threshold, which does not match the inference procedure and tests used in the paper. Could the rationale of this approach be described in the paper.

3. "All Bayesian analyses were conducted in JASP, using default priors.". There was previously some text which described the priors but this was removed. A reproducible analysis script is not available and the JASP version was not given. The default priors in JASP could be subject to change, and hence reproducibility of the analyses would be compromised. Therefore could some information on the priors be provided in the manuscript.

4. "A Bayesian one-way between-subjects ANOVA confirmed there were no differences between no-shift and shift conditions, F(1, 108) = .02, BF01 = 4.91." The word significant was removed in revisions. I understand the reason for this change since a NHST test was not performed. However, there was a numerical difference so this statement is incorrect. Similar comments apply to the results on p. 25 and perhaps elsewhere, e.g., "a Bayesian independent-samples t-test confirmed there was no difference between conditions, t(98) = .28, BF01 = 4.58". The description of the analysis on p.14 describes ranges of Bayes Factors for which qualitative labels for evidence are assigned based on conventions established in the literature. However, the inferential process that leads to the conclusion of "no difference" based on a BF value is not described, and the pre-registration does not articulate this either. In frequentist analysis the phrase "statistically significant difference" is used to qualify such statements, but such terminology to indicate qualification appropriate to Bayesian hypothesis testing is (mostly) absent here. Throughout, could results be described in a way that it is more clear as to what inference is being performed (with explanation of the inference procedure earlier on if necessary), and to qualify statements the results, i.e., it would seem more appropriate to talk about evidence for effects, or more how the Bayes Factor favours one model over another, rather than categorical statements of a presence/absence of an effect.

5. I apologise but I am still confused on the use of the ANOVA. On page 15 where this ANOVA is described, the means for three conditions were provided (i.e. including the control), but the following statement of a description of just a difference between two conditions (no-shift and shift), which would result from an ANOVA with two levels or a t-test. The following analysis on page 16 again described means for the control conditions along with shift and no-shift conditions, but here t-tests were used. My confusion here is that it does not appear that the control condition was included in any statistical analyses, and therefore I was unclear on the purpose of having it. Could this be explained in the text.

6. Page 26 reports Bayesian chi-square tests. There is not sufficient detail to allow reproducibility of these analyses in the manuscript. Could more explanation be provided, including priors used.

7. I did wonder why the approach taken in Experiment 2 was not used in Experiment 1 - an effect was found in the opposite direction in both cases but only Experiment 2 was a one-tailed test performed. I presume that the reason was that that in Experiment 2 there was weak evidence for the alternative hypothesis, and moderate evidence for the null in Experiment 1. But I wondered how the results of Experiment 1 would change (i.e. make the null findings more equivocal). I wonder if this was considered, and if it might be helpful, but I am not suggesting this is necessary.

     8. **Sensitivity to small effect sizes:**

Previously I asked for more consideration of design to detect small effect sizes. The response letter stated:
*"We have conducted a sensitivity analysis (using G*Power) for an independent samples t-test which found that our sample was large enough to detect an effect size of f = .27." *
The purpose of a sensitivity analysis would be to see the range of possible effect sizes that your design had sensitivity to detect. That you had a good power (in frequentist terms) to detect medium to large effects (i.e. f > .2) was not the issue being raised. A further sensitivity analysis was provided in the response editor letter compares H0 with f = 0 with H1 with f = 2 (d = .4). This is comparing a null effect with a medium-ish effect. The point I wanted to see addressed or discussed was the sensitivity of your design to smaller effects.
*"Additionally, in Experiment 1 the numeric effect between no-shift and shift conditions was in the opposite direction to our hypothesis, which speaks against a false-negative." *
Three possibilities can be considered to exist here. There is a true effect in the opposite direction to that predicted, a true null effect, or an true effect in the predicted direction. While it would be unlikely to find a Type S error (i.e., an error of the wrong sign) with a large effect in the predicted direction, the probability of this occurring with a smaller effect is non-negligible. The balance of evidence may favour one (or two) of these accounts over the other, but there does not seem to be very strong evidence to rule out any of these possibilities.Note, in Experiment 1, the difference in conditions appeared to be around d = .2, based on the difference in the means and an approximate calculation of the SD. Like Experiment 2, the effect was in the opposite direction as predicted. The BF01 was 3.02. A Bayes Factor provides a continuous measure, but this value is very close to the 3 threshold by some conventions the evidence could be described as anecdotal. Although numerical differences are highlighted in the text, the conclusions sometimes steer towards a more categorical interpretation, e.g. wordings such as "no effect, e.g. " no measurable impact", "failed to find an effect". Yet, with a frequentist analysis, the power would be approximately .18 for a similar sized effect (i.e., in r power.t.test(n=54, d=.2, type="two.sample"). The actual power is unknown, but my point is that a Bayesian analysis cannot overcome the same issue that a frequentist analysis has with detecting small effects with a limited sample size - in Bayesian terms, the probability of finding convincing evidence or misleading evidence (based on some threshold of evidence, e.g.  Schönbrodt & Wagenmakers, 2018).It seems to me that the evidence seems pretty convincing that there is not a medium or large effect, but the evidence for ruling out a small effect is not convincing. And based on the results of Experiment 2, a small effect in the opposite direction as predicted in Experiment 1 does seem possible. I do note that that in terms of your conclusions, either a null effect or an effect in the opposite direction in Experiment 1 does not change your substantive conclusions, hence this may be considered a minor point, and it is quite a generic one - i.e. most psychology designs use sample sizes in a range that make it difficult to detect small effects. Consequently, I don't think a sensitivity analysis is necessary in a revision, but this point can easily be addressed by more judicious use of language to describe evidence for effects in the results (as mentioned in comments above) and corresponding language to match the description of results in the discussion, and what can and cannot be plausibly ruled out.
---

## [Author Response · Author response to Decision Letter 1]

14 Jun 2022

See attached response to reviewers.

---

## [Editor Report · Decision Letter 2]

4 Jul 2022

Examining the role of information integration in the continued influence effect using an event segmentation approach

PONE-D-21-32228R2

Dear Dr. Sanderson,

We’re pleased to inform you that your manuscript has been judged scientifically suitable for publication and will be formally accepted for publication once it meets all outstanding technical requirements.

Kind regards,

Shane Lindsay

Academic Editor

PLOS ONE

---

## [Editor Report · Acceptance letter]

8 Jul 2022

PONE-D-21-32228R2 

Examining the role of information integration in the continued influence effect using an event segmentation approach 

Dear Dr. Sanderson:

I'm pleased to inform you that your manuscript has been deemed suitable for publication in PLOS ONE. Congratulations! Your manuscript is now with our production department. 

Kind regards, 

on behalf of

Dr. Shane Lindsay 

Academic Editor

PLOS ONE